

# Carnivore coexistence without competition: giant otters are more nocturnal around dens than sympatric neotropical otters

Darren Norris[1,2] and Fernanda Michalski[2,3,4]

[1] Environmental Sciences, Federal University of Amapá, Macpá, Brazil
[2] Ecology and Conservation of Amazonian Vertebrates Research Group, Federal University of Amapá, Macapá, Amapá, Brazil
[3] Postgraduate Programme in Tropical Biodiversity, Federal University of Amapá, Macapa, Brazil
[4] Pro-Carnivores Institute, Atibaia, São Paulo, Brazil

## ABSTRACT

Nocturnal activity of tropical otters is rarely reported. To date no studies have documented den use by sympatric giant (*Pteronura brasiliensis*) and neotropical otters (*Lontra longicaudis*). We used camera-traps to monitor den use by sympatric otters along an equatorial Amazonian river. Camera-traps provided evidence that giant otters were more nocturnal around dens than sympatric neotropical otters. Nocturnal activity was recorded in 11% of giant otter photos ($n = 14$ of 125 photos), but was recorded only once for neotropical otters. Den use by giant and neotropical otters overlapped spatially and temporally but not concurrently. We hypothesize that previously reported nocturnal activity in neotropical otters is facilitated by the absence or low density of giant otters. Our results also underscore the need to use complementary techniques together with den counts for monitoring otters as sympatric species can use the same dens.

# INTRODUCTION

Competition between sympatric carnivores (Order Carnivora) has been widely reported (*Perera-Romero et al., 2021*; *Smith, Erb & Pauli, 2023*; *Srivathsa et al., 2023*); however, sympatric otter species appear to coexist without direct competition (*Kruuk, 2006*). Otters, therefore, provide an ideal group to examine how coexistence is facilitated among potential competitors. Coexistence among otters is thought to be facilitated by niche differentiation/partitioning arising from several factors including differences in diet, resource use, spatial distribution, and temporal activity patterns (*Kruuk, 2006*; *Kruuk et al., 1994*). While niche partitioning has been suggested to facilitate species coexistence, resource use and activity of sympatric tropical otter species remains poorly studied.

Dens are an important resource for many mammals using terrestrial habitats. Dens are burrows or lairs that serve myriad functions including places to raise offspring, feed and shelter. The same den may be used by multiple species at once, even those that are unlikely

Corresponding author
Darren Norris,
darren.norris@unifap.br

to peacefully coexist such as intraguild competitors, or predators and prey (*Dupuis-Désormeaux et al., 2023*; *Mori, Menchetti & Balestrieri, 2015*). Semi-aquatic otters use dens mainly to shelter, rest and feed young (*Kruuk, 2006*). The presence of dens is often recorded and used as an indirect sign for monitoring otter distribution and abundance; however, den use and den sharing remains rarely documented in otters. An example comes from South Africa, where the same den was used at different times by three different African clawless otters (*Arden-Clarke, 1986*). To our knowledge there are no studies documenting den use by sympatric otter species.

Camera-traps have been used to provide insight into the distribution, behavior, and activity of rare mammals including den use by otters (*Caravaggi et al., 2017*; *Findlay et al., 2017*). Nocturnal activity has been reported in several otter species (*Kruuk, 2006*; *Perrin & Carranza, 2000*); however, despite many decades of studies nocturnal activity has only recently been reported in giant and neotropical otters using camera-traps (*Duplaix, Evangelista & Rosas, 2015*; *Leuchtenberger et al., 2014*; *Rheingantz et al., 2016*; *Rheingantz, Santiago-Plata & Trinca, 2017*). Giant and neotropical otters are important components of Amazonian freshwater systems, but the escalating impact of human disturbances poses a growing threat to both species (*Duplaix, Evangelista & Rosas, 2015*; *Rheingantz, Santiago-Plata & Trinca, 2017*). Giant otters are the largest freshwater otter and live in groups (typically 3-9 individuals) and are therefore expected to dominate the smaller bodied solitary neotropical otter (*Carter & Rosas, 1997*; *Duplaix, Evangelista & Rosas, 2015*; *Rheingantz, Santiago-Plata & Trinca, 2017*). Considering the threats to both species across Amazonia, it is imperative to enhance our understanding of their behaviors and interactions. This knowledge is essential for informing effective protection and management policies aimed at safeguarding these valuable components of the ecosystem.

Although camera-traps have been used to monitor den activity of giant and neotropical otters, to our knowledge there are no camera-trap studies from the species in sympatry. Both species are thought to be predominantly diurnal (*Duplaix, Evangelista & Rosas, 2015*; *Rheingantz, Santiago-Plata & Trinca, 2017*), but studies using camera-traps also confirm that both species can be nocturnal. Nocturnal activity has been reported in giant otters from the Brazilian Pantanal (*Leuchtenberger et al., 2014*) and neotropical otters in Atlantic Forest (*Rheingantz et al., 2016*), Orinoco River savanna (*Garrote et al., 2020*) and Amazonian regions (*Michalski et al., 2021*). Previous studies with camera-traps suggest nocturnal activity by giant otters around dens could be in response to the need to defecate, prey availability or predation risk (*Leuchtenberger et al., 2014*). Nocturnal activity around dens by neotropical otters has been suggested to be a response to human disturbances, potential competitors that are predominantly diurnal such as giant otters or prey availability (*Garrote et al., 2020*; *Rheingantz et al., 2016*).

When sympatric, the species are thought to coexist based on factors including differences in diet (*Moraes et al., 2021*) and activity, with neotropical otters suggested to be possibly more crepuscular and nocturnal when sympatric with giant otter (*Duplaix, 1980*). Here we use camera-traps to monitor den use by sympatric otters and test for shared/concurrent den use and whether neotropical otters are nocturnal when sympatric with giant otters.

## METHODS

### Ethics statement

This study used data from non-invasive field observations and did not involve direct contact or interactions with animals. Fieldwork was conducted under research permit numbers IBAMA/SISBIO 69342 to FM, issued by the Instituto Chico Mendes de Conservação da Biodiversidade (ICMBio). Cameras were directed towards dens and no photographs of people were taken.

### Den monitoring

Otter dens were monitored using camera-traps along 12.8 km of the Falsino River, Amapá State, Brazil (Fig. 1). Commercial fishing is not permitted along this river, which is bordered by closed canopy rainforest and protected areas on both banks and has only six resident families. Both otter species occurred historically along this river and have been monitored since 2011 (*Michalski et al., 2012*, *2021*; *Norris & Michalski, 2023*). Five dens were monitored simultaneously using camera-traps during the low river season from August to November 2021 (Supplemental Information S1, further details of the study area are presented in *Michalski et al. (2021)*). Dens were located approximately 16–29 km upstream from the nearest house and 75–88 km upstream from the nearest town (Porto Grande, with a resident population of less than 18 thousand (*Brazilian Institute of Geography and Statistics (IBGE), 2022*)). All dens (coded D1 to D5 along a downstream-upstream sequence) were dug into the river bank close to the river's edge (mean, range = 157 cm 60–330 cm) and were only accessible to otters during low river levels. Active dens (Supplemental Information S1) were located and selected based on results from previous studies (*Oliveira, Norris & Michalski, 2015*; *Togura, Norris & Michalski, 2014*) and from evidence of recent activity *i.e.*, clean den openings, fresh dirt and fresh otter tracks (*Groenendijk et al., 2005*).

A single camera-trap was installed close to the entrance of each den (95–193 cm, Supplemental Informations S1 and S2). Due to the proximity to water and the width of the river (>50 m) it was not possible to place multiple cameras at distances both near and far from the same den entrance as has been used for the Eurasian otter (*Findlay et al., 2017*). As cameras were close to the dens the field of view was limited and it was rarely possible to identify individuals or sex from photos.

### Data analysis

To compare activity between species we identified independent photos. This was done by selecting the first of consecutive photos of the same species at the same den within a 30-min interval. Although this reduced sample sizes compared with other approaches *e.g.*, identifying when animals leave the field of view (*Rowcliffe et al., 2014*), we chose this 30-min interval for two reasons. Firstly, it enabled us to reduce the influence of repeated photos of the group living giant otters, which could generate bias in comparisons of activity with the more solitary neotropical otters. And secondly it is also a widely adopted approach used for camera-trap studies and facilitates comparison with previous studies (*Leuchtenberger et al., 2014*; *Rheingantz et al., 2016*).
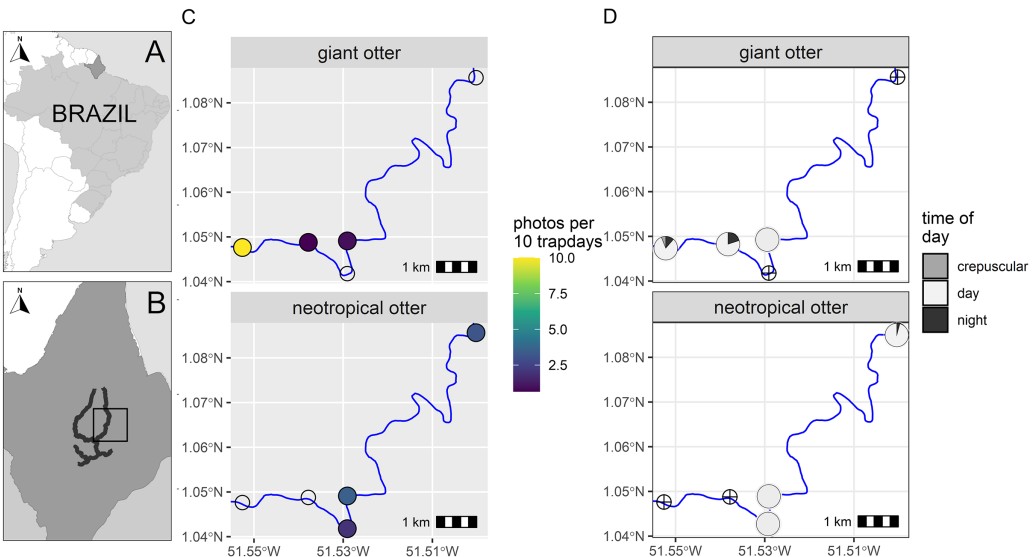

**Figure 1 Study area and otter activity.** (A) Location of the study area in South America and (B) the Brazilian State of Amapá. Maps showing locations of camera-traps at five otter dens along the river (solid line) with (C) detection rate and (D) diel activity of giant and neotropical otters. Circles with crosses show den locations where species were not recorded. Detections are independent photos per 10 trapdays. The proportion of diel activity at each den is shown among crepuscular (civil twilight), diurnal and nocturnal periods. Administrative boundaries and river lines from Natural Earth under a CC0 (public domain) license (https://www.naturalearthdata.com/downloads/).

To identify the relative frequency of den use, we identified den entrance events. As giant otters were most frequently recorded, we used data from this species to establish a 22.6 min threshold for distinct den events (Supplemental Information S3). As it is also likely that both species may use different dens/den entrances interchangeably it was not possible to consistently determine how long was spent inside or outside of dens. The chronology of den events did enable us to identify when otters remained overnight, and the proportion of overnight stays was used to complement the analysis of independent photos. While this is likely to be a minimum estimate as we did not monitor all entrances, it nonetheless allows for comparisons among dens and between species.

Activity levels and overlap between species were derived from kernel density estimates (*Rowcliffe et al., 2014*). Kernel density estimation is a non-parametric method used to estimate the probability density function of a continuous variable. This technique is applied to camera-trap data by fitting a flexible circular distribution to the detection times (when cameras were triggered by a given species). This distribution describes the underlying activity schedule and can be used to calculate the overall proportion of time active (*Ridout & Linkie, 2009*; *Rowcliffe et al., 2014*). Comparisons between species were made using functions available in R package activity (*Rowcliffe, 2023*). To test for significant differences in the distribution of activity *P* values were calculated from a randomisation test for the probability that two sets of circular observations came from the same distribution. Activity levels (proportion of the time active over 24H diel cycle) were compared using a Wald test for the statistical difference between activity level estimates.

## RESULTS

From a total of 88 days and 396 camera-trap days (range 77–88 camera trap days per camera, Supplemental Information S2), we obtained 192 independent photos of otters (*n* = 125 and *n* = 67 giant and neotropical photos respectively, Supplemental Information S2). Overall, the dens were used in a similar proportion by both species, with cameras recording den activity on 42% and 47% of days monitored (*n* = 37 and 41 daily cycles for which we had at least one record of giant and neotropical otters respectively). Giant otters were more active nocturnally than neotropical otters (Fig. 1). Nocturnal activity by giant otters was recorded on 27% of days with photos (*n* = 10 of 37 days) and from 11% of independent photos (*n* = 14 of 125 photos). In contrast, nocturnal activity by neotropical otters was recorded only once. Crepuscular activity by giant otters was recorded on 32% of days with photos (*n* = 12 of 37 days) and 11% of independent photos (*n* = 14 of 125 photos). All crepuscular activity by giant otters was recorded at a single den (D1), which was the only den where young were recorded. No crepuscular activity was recorded by neotropical otters (Fig. 1).

There was spatial separation in den use, with giant otters most frequently recorded at the most downstream den (D1) and neotropical otters at dens at least 3 km upstream (Fig. 1, Supplemental Information S2). Giant otters were recorded during consecutive days (five sequences of 3 to 5 days) and overnighted on 13 occasions at D1. Neotropical otters had fewer consecutive days at dens and were rarely recorded during three or more consecutive days at the same den (3 and 4 consecutive days at dens 4 and 5 respectively). The chronology of den events provided no evidence of neotropical otters overnighting at any of the dens.

Use of dens by both species did overlap spatially but not concurrently. Both species were recorded at D4 on 3 October. First a single adult giant otter entered and left D4 at 10:30H. Subsequently an adult neotropical otter entered and left this den in the late afternoon (17:04–17:05H). This same den was visited separately by both species on 26 days (*n* = 5 and *n* = 21 days for giant and neotropical otters respectively). Visits by both species to this den were usually short (less than 2 min between entering and leaving) and by a single adult. At least two adults of each species used this den (Supplemental Information S4), as two adults were recorded entering and leaving together on two separate days for each species. Only neotropical otters were recorded scent marking at this den. Scent marking by rubbing on the ground occurred on three different days (adult male on two days, while it was not possible to identify the gender of the adult on the other day).

The timing of den use differed between species. Comparison of Kernel density estimates showed significant inter-specific differences in the distribution of activity, yet there was substantial (>50%) overlap in timing of den use by both species (Fig. 2). Giant otters tended to be more active around dens (activity level 0.49 and 0.39 for giant and neotropical otters respectively), but this difference was not significant (Wald test *P* = 0.105). Giant otter den activity showed clear peaks (Fig. 2). On average giant otters left dens early morning (6H), returning around midday, leaving again around 13H–14H and entering at the end of the day (18H). No clear peaks were recorded for neotropical otter den use, but

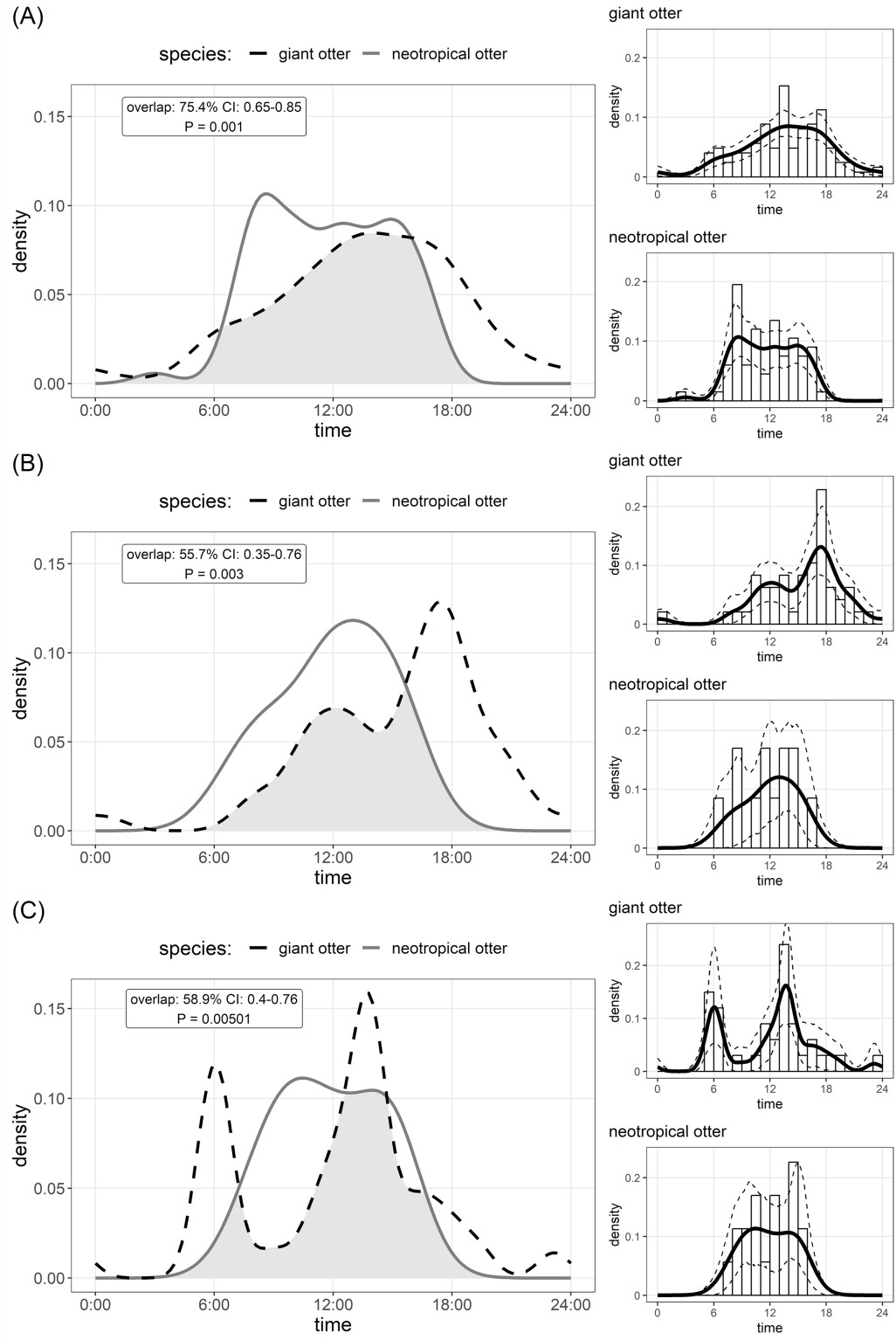

**Figure 2 Den activity of sympatric giant and neotropical otters.** Kernel density estimates and overlap of (A) overall activity, (B) entering dens, (C) leaving dens. The coefficient of overlap ranges from 0—no overlap—to 1—complete overlap. *P* values are from a randomisation test for the probability that two sets of circular observations come from the same distribution. Insets show distribution of independent photos with kernel density estimates and confidence intervals for each species.

den activity tended to be concentrated between 8H and 16H, with activity declining after 15H (Fig. 2).

## DISCUSSION

As far as we are aware our study is the first to evaluate den use by sympatric otters in the neotropics. We found that den use by both species overlapped spatially but not concurrently. Contrary to expectations, we also found that giant otters were more active at night than neotropical otters. Our findings provide an important baseline for studies to evaluate impacts on species activity patterns caused by anthropogenic changes.

During the low river season otters are restricted to the larger perennial rivers and access to resources including dens close to flowing rivers is therefore limited for both species. We suggest the observed patterns can be explained due to combination of den characteristics, den location and use by the species. The shared den had an entrance that was large enough for giant otters, but was far enough away from D1 so that it was visited infrequently by giant otters. In contrast, the entrance of D3 was likely too small for giant otters, which were never recorded at this den. The neotropical otters were not photographed with young at any den, whereas giant otters were found in a group with young exclusively at D1. Social organization and reproductive state could therefore be important factors in den activity, including how otters use dens for both shelter and feeding. D1 was the most frequently recorded location for giant otters, while neotropical otters were never observed there (Fig. 1). Thus, although both species can overlap den use under some conditions, neotropical otters appear to avoid dens most intensively used by large giant otter groups (D1 had at least seven individuals).

Plasticity in otter activity is expected and has been reported in different species (*Kruuk, 2006*). Semi-aquatic otters have adaptations that are likely to facilitate nocturnal activity, including eyes with a *tapetum lucidum* and well-developed iris sphincter muscles (*Ballard, Sivak & Howland, 1989*; *Kitchener, Meloro & Williams, 2017*). Although previous studies speculated that nocturnal activity in neotropical otters could be in response to anthropogenic disturbances (*Garrote et al., 2020*; *Rheingantz et al., 2016*), we suggest a non-mutually exclusive alternative hypothesis. We suggest that in the case of neotropical otters, patterns of no or little nocturnal activity could reflect the presence and proximity to giant otters. It would then follow that nocturnal activity of neotropical otters is facilitated by the absence or low density of giant otters. This would be true in cases such as the Atlantic Forest study area of *Rheingantz et al. (2016)*, with anthropogenic disturbances and no giant otters. Another factor that remains to be tested is the degree to which neotropical otter activity is associated with food availability. It could be that in our study area there is sufficient prey available, so neotropical otters do not need to be as active at night compared with Atlantic Forest study area, where main prey items including catfish and prawns were nocturnal/crepuscular (*Rheingantz et al., 2016*). Additionally, the study by *Garrote et al. (2020)* took place at a stretch of the Orinoco River with both commercial and recreational fishing. More intensive fishing activities could reduce prey availability for neotropical otters. If fewer prey were available then neotropical otters would need to be more active at

night compared with our study river, which is relatively far for the nearest town and where there are few houses and no commercial fishing.

A previous study rejected the hypothesis of giant otters influencing neotropical otter activity due to the absence of direct competition and as both species are predominantly diurnal (*Rheingantz et al., 2016*). However, our results provide new insight into the spatio-temporal relationships of these species in sympatry. Although we could not find reports of direct competition between otter species, direct competition of otters with other members of the same mustelid guild has been documented *e.g.*, interference competition with European otter (*Lutra lutra*) stealing fish from mink (*Mustela vison*) (*Bonesi, Dunstone & O'Connell, 2000*). Additionally giant otter groups are known to aggressively defend territories and infanticide has also been reported (*Carter & Rosas, 1997*; *Duplaix, Evangelista & Rosas, 2015*). Neotropical otters could therefore be expected to avoid nocturnal activity in regions where giant otters are active at night to reduce the chance of agnostic interspecific interactions.

Although neotropical otters have adaptations that improve the receptivity of eyes at night (*e.g.*, *tapetum lucidum*), their vision is still likely impaired at night as their eyes are quite small (*Larivière, 1999*). With small eyes neotropical otters are likely to be relatively short sighted and therefore less able to process visual cues at night. Avoiding nocturnal activity in areas where giant otters are active at night could therefore be a mechanism whereby neotropical otters reduce the chances of encounters when they would be less able to safely avoid or flee giant otters. Previous studies along the same rivers found diurnal river use by neotropical otters increased twofold in regions most intensively used by humans (*Norris & Michalski, 2023*). This finding does not support the hypothesis that neotropical otters reduce diurnal activity in regions with high levels of anthropogenic use. We suggest that the most likely explanation is that nocturnal activity previously reported in neotropical otters is at least in part facilitated by the absence/low density of giant otters.

Overall, neotropical otters tended to leave later in the morning and enter earlier in the afternoon compared with giant otters. A previous study quantifying river based activity found no evidence of direct interactions between giant and neotropical otters (*Norris & Michalski, 2023*). Indeed, *Norris & Michalski (2023)* found river use by giant and neotropical otters was separated temporally (median time difference 3.0 h) and spatially (median distance between species 12.5 km). However, results from this earlier study focused only on daytime activity and did not evaluate nocturnal activity of either species. It is likely that neotropical otters can obtain food resources in our study area without extending activity into the night. Diurnal activity of neotropical otters in our study area also included times (*e.g.*, midday) when giant otters are less active (*Duplaix, 1980*; *Leuchtenberger et al., 2014*; *Norris & Michalski, 2023*). Thereby reducing temporal overlap with giant otters.

Our results demonstrate that camera-traps do not appear to influence species activity, as both species were recorded using dens throughout the study period. This supports previous studies, which used camera-traps to monitor otter dens, reinforcing that long term monitoring in front of dens is a technique that can be applied in future studies to monitor otter species that are especially elusive and hard to monitor through visual

observation in densely forested areas such as Amazonian forests. Our results also highlight that the use of indirect sign such as den counts for otter species must be accompanied with complementary techniques as sympatric species can use the same dens.

## ACKNOWLEDGEMENTS
We are grateful to Alvino Pantoja Leal for his invaluable assistance during fieldwork.

### Funding
This work was funded by the Conservation, Food & Health Foundation. Fernanda Michalski receives a productivity scholarship from the Brazilian National Council for Scientific and Technological Development (CNPq – Process 310573/2021-1). The funders had no role in study design, data collection and analysis, decision to publish, or preparation of the manuscript.

### Grant Disclosures
The following grant information was disclosed by the authors:
Conservation, Food & Health Foundation.
Brazilian National Council for Scientific and Technological Development: CNPq – Process 310573/2021-1.

### Competing Interests
Fernanda Michalski is associated with Pro-Carnivores Institute. Fernanda Michalski is an Academic Editor for PeerJ. Darren Norris is an Academic Editor for PeerJ. The authors declare there are no competing interests.

### Author Contributions
- Darren Norris conceived and designed the experiments, analyzed the data, prepared figures and/or tables, authored or reviewed drafts of the article, and approved the final draft.
- Fernanda Michalski conceived and designed the experiments, performed the experiments, authored or reviewed drafts of the article, and approved the final draft.

### Animal Ethics
The following information was supplied relating to ethical approvals (i.e., approving body and any reference numbers):
  Fieldwork was conducted under research permit numbers IBAMA/SISBIO 69342 to FM, issued by the Instituto Chico Mendes de Conservação da Biodiversidade (ICMBio). Cameras were directed towards dens and no photographs of people were taken.

### Data Availability
  The raw data used in the analysis are available in the Supplemental File.

## Supplemental Information

Supplemental information for this article can be found online at http://dx.doi.org/10.7717/peerj.17244#supplemental-information.

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
