# Peer review of "Carnivore coexistence without competition: giant otters are more nocturnal around dens than sympatric neotropical otters"

_PeerJ, doi:10.7717/peerj.17244_

## Round 0.1 · original submission · Minor Revisions

Three expert reviewers all had positive impressions of your work and have provided very minor suggestions for revision. This is a novel and straightforward study. I have some additional minor requests, however.

Line 69, do you mean “compared to when they are not sympatric with giant otters?” Please write this out in full. Whenever you make comparison statements, make the comparison very explicit.

Please explain kernel density estimates (line 117) for naïve readers.

Minor issues:

In the abstract, the sentence beginning “Whereas…” is incomplete. Please combine with the preceding sentence. Similarly for lines 65-67.

Please use ; before however (e.g., line 31 and 43 and check throughout). Always place , after however (e.g., line 49).

Please place a comma around “therefore” (line 32.

Please use commas after clauses such as “when sympatric.. on line 68. And after “den use” on line 109. Check throughout.

Line 42, it would be more precise to say “the presence” or “use” of dens is recorded. Place “a” before “indirect sign.”

Line 115, do not use contractions.

·

Basic reporting

1. Basic Reporting: This article is well-written in clear, unambiguous English. The references are up to date and list the appropriate background data to assertions made in the text. The raw data provided supports the results in the text and the figures/illustrations. The length of the article is appropriate for the topic and the supporting results.

Experimental design

2. Experimental Design: While other articles have been published on den use in giant otters and Neotropical otters, this article examines den use overlap by both species in the dry season over time which is useful. The level of detail provided as background information to the investigation is clear and complete. It will be straightforward to duplicate this investigation in other areas where these two otter species are sympatric, an important aspect in conservation planning for both these species.

Validity of the findings

3. Validity of the Findings: Findings based on camera trapping data are valid as supporting data are collected on the SIM card for each photograph. The interpretation of the data and the conclusions are clear, detailed, and well stated.

Additional comments

I would like to congratulate the authors for filling some gaps in our knowledge of the relationship between these two sympatric species.

Reviewer 2 ·

Basic reporting

This ms uses clear and unambiguous English throughout. I believe the literature is appropriate except that their statement on line 32 needs a reference. I believe the authors gave a good introduction that puts the study in context. The figures are professional and the discussion does not stray from the results.

Experimental design

this study used camera trap sightings to describe den use by 2 sympatric species of South American otters. I see no flaw with the design.

Validity of the findings

the findings are straight forward and the interpretation is valid. Data is relatively simple but used well and the conclusions are supported.

Additional comments

Just one minor suggestion: Line 32-please provide references-this is a statement that needs support.

Reviewer 3 ·

Basic reporting

The article is weel-written, has recent references and enough background/context of the work.
The article has good artcile strcture, size and it is written correcly.
The figure 1 must include also an arrow with the north. About figure 2, I suggest to use the same scale in all graphs of the figure, to stardarize, in special the "Y" axis.

Experimental design

The design is well-defined. The authors mention 30 minutes as independence criteria. Considering the entrance/exit of the den, actitives of both species can be inffluenced by previous one. So, I suggest to use 1-h as independence criteria.
I suggest ith lowest and highest values, and the effort of each camera (in cameratrap days).
The research question is well-defined, is relevant and it fills partially an identified knowledge of gap.

Validity of the findings

The findings are interesting and robust, showing that in Amapá the giant otters and Neotropical Otter can use the same dens, and in this area Neotropical otter has only diurnal activity ,and giant otter can show also nocturnal activity. This last finding was previously published by Leuchtenberger et al 2014. Other interesting finding was the temporal and spatial overlpa in the use of the dens between both species. However, the statement that the nocturnal behaviour of Neotropical Otter is rare is not true, as shown at least in Rheingantz et al. 2016, Carvalho-Jr 2007, Garrote et al 2020. Garrote et al and Rheingantz et al articles showed that in areas with historical presence of humans, otters can show nocturnal and crepuscular activity. In Garrote et al study, atlhough they have a small number of independent registers, they mention that in the area where they studied Neotropical Otter cooccur with Giant Otter, and the NO is nocturnal. Larriviere 1999, in his review, mention that NO can be nocturnal when affected by human activities. I suggest to include the population numbers of Falsino River and surroudings, to understand the level of human impact in the area.
Conclusion are mainly well stated, but the hypothesis that NO became diurnal to run from GO is speculative, as we do not have information of the same behaviour in different areas, and even in Pantanal, where Rheingantz et al shown NO activity in the crepuscular periods, although less than during the day i Pantanal, where the species coexists with GO.

---

## Round 0.2 · accepted · Accept

Thank you for your attention to the revisions requested in the first round of review.